# DQSGD: Dynamic Quantized Stochastic Gradient Descent for Communication-efficient Distributed Learning

## Abstract

Gradient quantization is widely adopted to mitigate communication costs in distributed learning systems. Existing gradient quantization algorithms often rely on design heuristics and/or empirical evidence to tune the quantization strategy for different learning problems. To the best of our knowledge, there is no theoretical framework characterizing the trade-off between communication cost and model accuracy under dynamic gradient quantization strategies. This paper addresses this issue by proposing a novel dynamic quantized SGD (DQSGD) framework, which enables us to optimize the quantization strategy for each gradient descent step by exploring the trade-off between communication cost and modeling error. In particular, we derive an upper bound, tight in some cases, of the modeling error for arbitrary dynamic quantization strategy. By minimizing this upper bound, we obtain an enhanced quantization algorithm with significantly improved modeling error under given communication overhead constraints. Besides, we show that our quantization scheme achieves a strengthened communication cost and model accuracy trade-off in a wide range of optimization models. Finally, through extensive experiments on large-scale computer vision and natural language processing tasks on CIFAR-10, CIFAR-100, and AG-News datasets, respectively. we demonstrate that our quantization scheme significantly outperforms the state-of-the-art gradient quantization methods in terms of communication costs.

## 1 Introduction

Recently, with the booming of Artificial Intelligence (AI), 5G wireless communications, and Cyber-Physical Systems (CPS), distributed learning plays an increasingly important role in improving the efficiency and accuracy of learning, scaling to a large input data size, and bridging different wireless computing resources (Dean et al., 2012; Bekkerman et al., 2011; Chilimbi et al., 2014; Chaturapruek et al., 2015; Zhu et al., 2020; Mills et al., 2019). Distributed Stochastic Gradient Descent (SGD) is the core in a vast majority of distributed learning algorithms (e.g., various distributed deep neural networks), where distributed nodes calculate local gradients and an aggregated gradient is achieved via communication among distributed nodes and/or a parameter server.

However, due to limited bandwidth in practical networks, communication overhead for transferring gradients often becomes the performance bottleneck. Several approaches towards communication-efficient distributed learning have been proposed, including compressing gradients (Stich et al., 2018; Alistarh et al., 2017) or updating local models less frequently (McMahan et al., 2017). Gradient quantization reduces the communication overhead by using few bits to approximate the original real value, which is considered to be one of the most effective approaches to reduce communication overhead (Seide et al., 2014; Alistarh et al., 2017; Bernstein et al., 2018; Wu et al., 2018; Suresh et al., 2017). The lossy quantization inevitably brings in gradient noise, which will affect the convergence of the model. Hence, a key question is how to effectively select the number of quantization bits to balance the trade-off between the communication cost and its convergence performance.

Existing algorithms often quantize parameters into a fixed number of bits, which is shown to be inefficient in balancing the communication-convergence trade-off (Seide et al., 2014; Alistarh et al., 2017; Bernstein et al., 2018). An efficient scheme should be able to dynamically adjust the number

of quantized bits according to the state of current learning model in each gradient descent step to balance the communication overhead and model accuracy. Several studies try to construct adaptive quantization schemes through design heuristics and/or empirical evidence. However, they do not come up with a solid theoretical analysis (Guo et al., 2020; Cui et al., 2018; Oland & Raj, 2015), which even results in contradicted conclusions. More specifically, MQGrad (Cui et al., 2018) and AdaQS (Guo et al., 2020) suggest using few quantization bits in early epochs and gradually increase the number of bits in later epochs; while the scheme proposed by Anders (Oland & Raj, 2015) states that more quantization bits should be used for the gradient with larger root-mean-squared (RMS) value, choosing to use more bits in the early training stage and fewer bits in the later stage. One of this paper's key contributions is to develop a theoretical framework to crystallize the design tradeoff in dynamic gradient quantization and settle this contradiction.

In this paper, we propose a novel dynamic quantized SGD (DQSGD) framework for minimizing communication overhead in distributed learning while maintaining the desired learning accuracy. We study this dynamic quantization problem in both the strongly convex and the non-convex optimization frameworks. In the strongly convex optimization framework, we first derive an upper bound on the difference (that we term the strongly convex convergence error) between the loss after $N$ iterations and the optimal loss to characterize the strongly convex convergence error caused by sampling, limited iteration steps, and quantization. In addition, we find some particular cases and prove the tightness for this upper bound on part of the convergence error caused by quantization. In the non-convex optimization framework, we derive an upper bound on the mean square of gradient norms at every iteration step, which is termed the non-convex convergence error. Based on the above theoretical analysis, we design a dynamic quantization algorithm by minimizing the strongly convex/non-convex convergence error bound under communication cost constraints. Our dynamic quantization algorithm is able to adjust the number of quantization bits adaptively by taking into account the norm of gradients, the communication budget, and the remaining number of iterations. We validate our theoretical analysis through extensive experiments on large-scale Computer Vision (CV) and Natural Language Processing (NLP) tasks, including image classification tasks on CIFAR-10 and CIFAR-100 and text classification tasks on AG-News. Numerical results show that our proposed DQSGD significantly outperforms the baseline quantization methods.

To summarize, our key contributions are as follows:

• We propose a novel framework to characterize the trade-off between communication cost and modeling error by dynamically quantizing gradients in the distributed learning.

• We derive an upper bound on the convergence error for strongly convex objectives and non-convex objectives. The upper bound is shown to be optimal in particular cases.

• We develop a dynamic quantization SGD strategy, which is shown to achieve a smaller convergence error upper bound compared with fixed-bit quantization methods.

• We validate the proposed DQSGD on a variety of real world datasets and machine learning models, demonstrating that our proposed DQSGD significantly outperforms state-of-the-art gradient quantization methods in terms of mitigating communication costs.

## 2 RELATED WORK

To solve large scale machine learning problems, distributed SGD methods have attracted a wide attention (Dean et al., 2012; Bekkerman et al., 2011; Chilimbi et al., 2014; Chaturapruek et al., 2015). To mitigate the communication bottleneck in distributed SGD, gradient quantization has been investigated. 1BitSGD uses 1 bit to quantize each dimension of the gradients and achieves the desired goal in speech recognition applications (Seide et al., 2014). TernGrad quantizes gradients to ternary levels $\{-1, 0, 1\}$ to reduce the communication overhead (Wen et al., 2017). Furthermore, QSGD is considered in a family of compression schemes that use a fixed number of bits to quantize gradients, allowing the user to smoothly trade-off communication and convergence time (Alistarh et al., 2017). However, these fixed-bit quantization methods may not be efficient in communication. To further reduce the communication overhead, some empirical studies began to dynamically adjust the quantization bits according to current model parameters in the training process, such as the gradient's mean to standard deviation ratio (Guo et al., 2020), the training loss (Cui et al., 2018), gradient's root-mean-squared value (Oland & Raj, 2015). Though these empirical heuristics of adaptive quan-

tization methods show good performance in some certain tasks, their imprecise conjectures and the lack of theoretical guidelines in the conjecture framework have limited their generalization to a broad range of machine learning models/tasks.

## 3 PROBLEM FORMULATION

We consider to minimize the objective function $F : \mathbb{R}^d \to \mathbb{R}$ with parameter $\mathbf{x}$

$$\min_{\mathbf{x} \in \mathbb{R}^d} F(\mathbf{x}) = \mathbb{E}_{\xi \sim D}[l(\mathbf{x}; \xi)], \tag{1}$$

where the data point $\xi$ is generated from an unknown distribution $D$, and a loss function $l(\mathbf{x}; \xi)$ measures the loss of the model $\mathbf{x}$ at data point $\xi$. Vanilla gradient descent (GD) will solve this problem by updating model parameters via iterations $\mathbf{x}^{(n+1)} = \mathbf{x}^{(n)} - \eta \nabla F(\mathbf{x}^{(n)})$, where $\mathbf{x}^{(n)}$ is the model parameter at iteration $n$; $\eta$ is the learning rate; $\nabla F(\mathbf{x}^{(n)})$ is the gradient of $F(\mathbf{x}^{(n)})$. A modification to the GD scheme, minibatch SGD, uses mini-batches of random samples with size $K$, $A_K = \{\xi_0, ..., \xi_{K-1}\}$, to calculate the stochastic gradient $g(\mathbf{x}) = 1/K \sum_{i=0}^{K-1} \nabla l(\mathbf{x}; \xi_i)$.

In distributed learning, to reduce the communication overhead, we consider to quantize the minibach stochastic gradients:

$$\mathbf{x}^{(n+1)} = \mathbf{x}^{(n)} - \eta Q_{s_n}[g(\mathbf{x}^{(n)})], \tag{2}$$

where $Q_{s_n}[\cdot]$ is the quantization operation that works on each dimension of $g(\mathbf{x}^{(n)})$. The $i$-th component of the stochastic gradient vector $g$ is quantized as

$$Q_s(g_i) = \|g\|_p \cdot \text{sgn}(g_i) \cdot \zeta(g_i, s), \tag{3}$$

where $\|g\|_p$ is the $l_p$ norm of $g$; $\text{sgn}(g_i) = \{+1, -1\}$ is the sign of $g_i$; $s$ is the quantization level; and $\zeta(g_i, s)$ is an unbiased stochastic function that maps scalar $|g_i|/\|g\|_p$ to one of the values in set $\{0, 1/s, 2/s, \ldots, s/s\}$: if $|g_i|/\|g\|_p \in [l/s, (l+1)/s]$, we have

$$\zeta(g_i, s) = \begin{cases} l/s, & \text{with probability } 1 - p, \\ (l+1)/s, & \text{with probability } p = s\dfrac{|g_i|}{\|g\|_p} - l. \end{cases} \tag{4}$$

Note that, the quantization level is roughly exponential to the number of quantized bits. If we use $B$ bits to quantize $g_i$, we will use one bit to represent its sign and the other $B - 1$ bits to represent $\zeta(g_i, s)$, thus resulting in a quantization level $s = 2^{B-1} - 1$. In total, we use $B_{pre} + dB$ bits for the gradient quantization at each iteration: a certain number of $B_{pre}$ bits of precision to construct $\|g\|_p$ and $dB$ bits to express the $d$ components of $g$.

Given a total number of training iterations $N$ and the overall communication budget $C$ to upload all stochastic gradients, we would like to design a gradient quantization scheme to maximize the learning performance. To measure the learning performance under gradient quantization, we follow the commonly adopted convex/non-convex-convergence error $\delta(F, N, C)$ (Alistarh et al., 2017):

$$\delta(F, N, C) = \begin{cases} F(\mathbf{x}^{(N)}, C) - F(\mathbf{x}^*, C), & \text{for strongly convex } F, \\ \dfrac{1}{N} \sum_{n=0}^{N-1} \|\nabla F(\mathbf{x}^{(n)})\|_2^2, & \text{for non-convex } F, \end{cases} \tag{5}$$

where $\mathbf{x}^*$ is the optimal point to minimize $F$. In general, this error $\delta(F, N, C)$ is hard to determine; instead, we aim to lower and upper bound this error and design corresponding quantization schemes.

## 4 DYNAMIC QUANTIZED SGD

In this part, we derive upper bounds on the strongly convex/non-convex convergence error $\delta(F, N, C)$ and lower bounds on the strongly convex-convergence error. By minimizing the upper bound on this convergence error, we propose the dynamic quantized SGD strategies for strongly convex and non-convex objective functions.

### 4.1 PRELIMINARIES

We first state some assumptions as follows.

**Assumption 1** (Smoothness). *The objective function $F(\mathbf{x})$ is L-smooth, if $\forall \mathbf{x}, \mathbf{y} \in \mathbb{R}^d$, $\|\nabla F(\mathbf{x}) - \nabla F(\mathbf{y})\|_2 \leqslant L\|\mathbf{x} - \mathbf{y}\|_2$.*

It implies that $\forall \mathbf{x}, \mathbf{y} \in \mathbb{R}^d$, we have

$$F(\mathbf{y}) \leq F(\mathbf{x}) + \nabla F(\mathbf{x})^{\mathrm{T}}(\mathbf{y} - \mathbf{x}) + \frac{L}{2}\|\mathbf{y} - \mathbf{x}\|_2^2 \tag{6}$$

$$\|\nabla F(\mathbf{x})\|_2^2 \leq 2L[F(\mathbf{x}) - F(\mathbf{x}^*)] \tag{7}$$

**Assumption 2** (Strongly convexity). *The objective function $F(\mathbf{x})$ is $\mu$-strongly convex, if $\exists \mu > 0$, $F(\mathbf{x}) - \frac{\mu}{2}\mathbf{x}^T\mathbf{x}$ is a convex function.*

From Assumption 2, we have: $\forall \mathbf{x}, \mathbf{y} \in \mathbb{R}^d$,

$$F(\mathbf{y}) \geq F(\mathbf{x}) + \nabla F(\mathbf{x})^{\mathrm{T}}(\mathbf{y} - \mathbf{x}) + \frac{\mu}{2}\|\mathbf{y} - \mathbf{x}\|_2^2 \tag{8}$$

**Assumption 3** (Variance bound). *The stochastic gradient oracle gives us an independent unbiased estimate $\nabla l(\mathbf{x}; \xi)$ with a bounded variance:*

$$\mathbb{E}_{\xi \sim D}[\nabla l(\mathbf{x}; \xi)] = \nabla F(\mathbf{x}), \tag{9}$$

$$\mathbb{E}_{\xi \sim D}[\|\nabla l(\mathbf{x}; \xi) - \nabla F(\mathbf{x})\|_2^2] \leq \sigma^2. \tag{10}$$

From Assumption 3, for the minibatch stochastic gradient $g(\mathbf{x}) = [\sum_{i=0}^{K-1} \nabla l(\mathbf{x}; \xi_i)]/K$, we have

$$\mathbb{E}_{\xi \sim D}[g(\mathbf{x})] = \nabla F(\mathbf{x}) \tag{11}$$

$$\mathbb{E}_{\xi \sim D}[\|g(\mathbf{x}; \xi)\|^2] \leq \|\nabla F(\mathbf{x})\|_2^2 + \sigma^2/K. \tag{12}$$

We have the relationship of gradients before and after quantization: $Q_s[g(\mathbf{x})] = g(\mathbf{x}) + \hat{\epsilon}$, where $\hat{\epsilon}$ represents the quantization noise, following the probability distribution that can be shown in Proposition 1. The proof of Proposition 1 is given in Appendix A.

**Proposition 1** (Quantization Noise magnitude). *For the stochastic gradient vector $g$, if the quantization level is $s$, then the $i$-th component of quantization noise follows as:*

$$p(\hat{\epsilon}_i) = \begin{cases} \dfrac{s}{\|g\|_p} - \dfrac{s^2}{\|g\|_p^2}\hat{\epsilon}_i, & 0 < \hat{\epsilon}_i \leq \dfrac{\|g\|_p}{s}, \\ \dfrac{s}{\|g\|_p} + \dfrac{s^2}{\|g\|_p^2}\hat{\epsilon}_i, & -\dfrac{\|g\|_p}{s} \leq \hat{\epsilon}_i \leq 0. \end{cases} \tag{13}$$

Following Proposition 1, we can get $\mathbb{E}_{\hat{\epsilon}_i}[Q_s[g]] = g$ and $\mathbb{E}_{\hat{\epsilon}_i}[\|Q_s[g] - g\|_2^2] = \dfrac{d}{6s^2}\|g\|_p^2$. This indicates that the quantization operation is unbiased, and the variance bound of $Q_s[g]$ is directly proportional to $\|g\|_p^2$ and inversely proportional to $s^2$, which means that gradients with a larger norm should be quantized using more bits to keep $\mathbb{E}[\|Q_s[g] - g\|_2^2]$ below a given noise level. Therefore, we have the following lemma to characterize the quantization noise $Q_s[g]$.

**Lemma 1.** *For the quantized gradient vector $Q_s[g]$, we have*

$$\mathbb{E}[Q_s[g]] = \nabla F(\mathbf{x}) \tag{14}$$

$$\mathbb{E}[\|Q_s[g]\|_2^2] \leq \|\nabla F(\mathbf{x})\|_2^2 + \frac{\sigma^2}{K} + \frac{d}{6s^2}\|g\|_p^2 \tag{15}$$

We can see that the noise various of $Q_s[g]$ contains two parts: the first part is the sampling noise $\dfrac{\sigma^2}{K}$, the second part is the quantization noise $\dfrac{d}{6s^2}\|g\|_p^2$.

## 4.2 Convergence error of strongly convex objectives

Firstly, we consider a strongly convex optimization problem. Putting the QSGD algorithm (2) on smooth, strongly convex functions yield the following result with proof given in Appendix B.

**Theorem 1** (Convergence Error Bound of Strongly Convex Objectives). *For the problem in Eq.* (1) *under Assumption 1 and Assumption 2 with initial parameter* $\mathbf{x}^{(0)}$*, using quantized gradients in Eq.* (2) *for iteration, we can upper and lower bound the convergence error by*

$$\mathbb{E}[F(\mathbf{x}^{(N)}) - F(\mathbf{x}^*)] \leq \alpha^N[F(\mathbf{x}^{(0)}) - F(\mathbf{x}^*)] + \frac{L\eta^2\sigma^2(1-\alpha^N)}{2K(1-\alpha)}$$
$$+ \frac{Ld\eta^2}{12}\sum_{n=0}^{N-1}\alpha^{N-1-n}\frac{1}{s_n^2}\|g(\mathbf{x}^{(n)})\|_p^2,$$

$$\mathbb{E}[F(\mathbf{x}^{(N)}) - F(\mathbf{x}^*)] \geq \beta^N[F(\mathbf{x}^{(0)}) - F(\mathbf{x}^*)] + \frac{\mu\eta^2\sigma^2(1-\beta^N)}{2K(1-\beta)}$$
$$+ \frac{\mu d\eta^2}{12}\sum_{n=0}^{N-1}\beta^{N-1-n}\frac{1}{s_n^2}\|g(\mathbf{x}^{(n)})\|_p^2,$$

where $\alpha = 1 - 2\mu\eta + L\mu\eta^2$, $\beta = 1 - 2L\eta + L\mu\eta^2$. The convergence error consists of three parts: the error of the gradient descent method, which which tends to 0 as the number of iterations $N$ increases and also depends on the learning rate $\eta$ (from the expression of $\alpha$, we can see that when $\eta \leq 1/L$, with the increase of $\eta$, $\alpha$ decrease, and the convergence rate of the model is accelerated); the sampling error, which can be reduced by increasing the batch size $K$ or decaying the learning rate; and the convergence error due to quantization, which we want to minimize. Note that there is a positive correlation between the upper bound of convergence error due to quantization and the variance of the quantization noise. The contribution of quantization noise to the error is larger at the late stage of training. Therefore, noise reduction helps improve the accuracy of the model. In other words, more quantization bits should be used in the later training period.

In addition, we can show that the upper and lower bound matches each other in some particular cases. As a simple example, we consider a quadratic problem: $F(\mathbf{x}) = \mathbf{x}^T\mathbf{H}\mathbf{x} + \mathbf{A}^T\mathbf{x} + B$, where the Hessian matrix is isotropic $\mathbf{H} = \lambda I$, $\mathbf{A} \in \mathbb{R}^d$ and $B$ is a constant. Clearly, $L = \mu$, so $\alpha = \beta$ and the upper is equal to the lower bound.

**Theorem 2** (Convergence Error of Quadratic Functions). *For a quadratic optimization problem* $F(\mathbf{x}) = \mathbf{x}^T\mathbf{H}\mathbf{x} + \mathbf{A}^T\mathbf{x} + B$*, we consider a Gaussian noise case*

$$\mathbf{x}^{(n+1)} = \mathbf{x}^{(n)} - \eta\nabla F(\mathbf{x}^{(n)}) - \eta\boldsymbol{\epsilon}^{(n)}, \boldsymbol{\epsilon}^{(n)} \sim \mathcal{N}(\mathbf{0}, \boldsymbol{\Sigma}(\mathbf{x}^{(n)})). \tag{16}$$

*We achieve*

$$\mathbb{E}[F(\mathbf{x}^{(N)}) - F(\mathbf{x}^*)] = \frac{1}{2}(\mathbf{x}^{(0)} - \mathbf{x}^*)^T(\boldsymbol{\rho}^N)^T\mathbf{H}\boldsymbol{\rho}^N(\mathbf{x}^{(0)} - \mathbf{x}^*)$$
$$+ \frac{\eta^2}{2}\sum_{n=0}^{N-1}\mathrm{Tr}[\boldsymbol{\rho}^{N-1-n}\boldsymbol{\Sigma}(\mathbf{x}^{(n)})\mathbf{H}(\boldsymbol{\rho}^{N-1-n})^T], \tag{17}$$

*where* $\rho = \mathbf{I} - \eta\mathbf{H}$ *and* $\mathbf{H}$ *is the Hessian matrix.*

Detailed proof is in Appendix C.

### 4.3 DQSGD FOR STRONGLY CONVEX OBJECTIVES

We will determine the dynamic quantization strategy by minimizing the upper bound of convergence error due to quantization. The optimization problem is:

$$\min_{B_n} \sum_{n=0}^{N-1} \alpha^{N-1-n} \frac{1}{(2^{B_n-1}-1)^2} \|g(\mathbf{x}^{(n)})\|_p^2,$$

$$\sum_{n=0}^{N-1} (dB_n + B_{pre}) = C.$$

By solving this optimization problem, we can get

$$B_n = \log_2 \left[ k\alpha^{(N-n)/2} \|g(\mathbf{x}^{(n)})\|_p + 1 \right] + 1, \tag{18}$$

where $k$ depends on the total communication overhead $C$, and $\alpha$ is related to the convergence rate of the model. The larger the total communication cost $C$ is, the greater $k$ is; the faster the model's convergence rate is, the smaller $\alpha$ is. In Appendix E, we prove that our scheme outperforms the fixed bits scheme in terms of the convergence error.

### 4.4 DQSGD FOR NON-CONVEX OBJECTIVES

In general, if we consider non-convex smooth objective functions, we can get the following theorem with proofs given in Appendix D.

**Theorem 3** (Convergence Error Bound of Non-Convex Objectives). *For the problem in Eq. (1) under Assumption 1, with initial parameter $\mathbf{x}^{(0)}$, using quantized gradients in Eq. (2) for iteration, we can upper bound the convergence error by*

$$\frac{1}{N} \sum_{n=0}^{N-1} \mathbb{E}[\|\nabla F(\mathbf{x}^{(n)})\|_2^2] \leq \frac{2}{2N\eta - LN\eta^2} [F(\mathbf{x}^{(0)}) - F(\mathbf{x}^*)] + \frac{L\eta\sigma^2}{(2-L\eta)K}$$
$$+ \frac{Ld\eta}{6(2-L\eta)N} \sum_{n=0}^{N-1} \frac{1}{s_n^2} \|g(\mathbf{x}^{(n)})\|_p^2. \tag{19}$$

Similarly, the convergence error consists of three parts: the error of the gradient descent method, which tends to 0 as the number of iterations $N$ increases; the sampling error, which can be reduced by increasing the batch size $K$ or decaying the learning rate; and the convergence error due to quantization, which we want to minimize. Thus, the optimization problem is:

$$\min_{B_n} \sum_{n=0}^{N-1} \frac{1}{s_n^2} \|g(\mathbf{x}^{(n)})\|_p^2$$

$$\sum_{n=0}^{N-1} (dB_n + B_{pre}) = C$$

By solving this optimization problem, we can get

$$B_n = \log_2 \left[ t\|g(\mathbf{x}^{(n)})\|_p + 1 \right] + 1, \tag{20}$$

where $t$ depends on the total communication overhead $C$. In Appendix E, we also give a detailed comparison of our scheme's the upper bound of convergence error compared with fixed-bit schemes.

### 4.5 DQSGD IN DISTRIBUTED LEARNING

Next, we consider the deployment of our proposed DQSGD algorithm in the distributed learning setting. We have a set of $W$ workers who proceed in synchronous steps, and each worker has a complete copy of the model. In each communication round, workers compute their local gradients and communicate gradients with the parameter server, while the server aggregates these gradients

from workers and updates the model parameters. If $\tilde{g}^l(\mathbf{x}^{(n)})$ is the quantized stochastic gradients in the $l$-th worker and $\mathbf{x}^{(n)}$ is the model parameter that the workers hold in iteration $n$, then the updated value of $\mathbf{x}$ by the end of this iteration is: $\mathbf{x}^{(n+1)} = \mathbf{x}^{(n)} + \eta \tilde{G}(\mathbf{x}^{(n)})$, where $\tilde{G}(\mathbf{x}^{(n)}) = \frac{1}{W} \sum_{l=1}^{W} \tilde{g}^l(\mathbf{x}^{(n)})$. The pseudocode is given in Algorithm 2 in Appendix E .

## 5 EXPERIMENTS

In this section, we conduct experiments on CV and NLP tasks on three datasets: AG-News (Zhang et al., 2015), CIFAR-10, and CIFAR-100 (Krizhevsky et al., 2009), to validate the effectiveness of our proposed DQSGD method. We use the testing accuracy to measure the learning performance and use the compression ratio to measure the communication cost. We compare our proposed DQSGD with the following baselines: SignSGD (Seide et al., 2014), TernGrad (Wen et al., 2017), QSGD (Alistarh et al., 2017), Adaptive (Oland & Raj, 2015), AdaQS (Guo et al., 2020). We conduct experiments for $W = 8$ workers and use canonical networks to evaluate the performance of different algorithms: BiLSTM on the text classification task on the AG-News dataset, Resnet18 on the image classification task on the CIFAR-10 dataset, and Resnet34 on the image classification task on the CIFAR-100 dataset. A detailed description of the three datasets, the baseline algorithms, and experimental setting is given in Appendix F.

**Test Accuracy vs Compression Ratio**. In Table 1, we compare the testing accuracy and compression ratio of different algorithms under different tasks. We can see that although SignSGD, TernGrad, QSGD (4 bits) have a compression ratio greater than 8, they cannot achieve more than 0.8895, 0.8545, 0.6840 test accuracy for AG-News, CIFAR-10, CIFAR-100 tasks, respectively. In contrast, QSGD (6 bits), Adaptive, AdaQS, and DQSGD can achieve more than 0.8986, 0.8785, 0.6939 test test accuracy. Among them, our proposed DQSGD can save communication cost by $4.11\% - 21.73\%$, $22.36\% - 25\%$, $11.89\% - 24.07\%$ than the other three algorithms.

Table 1: Accuracy vs. compression ratio.

|  | AG-News | | CIFAR-10 | | CIFAR-100 | |
|---|---|---|---|---|---|---|
|  | Top-1 Accuracy | Compression Ratio | Top-1 Accuracy | Compression Ratio | Top-1 Accuracy | Compression Ratio |
| Vanilla SGD | 0.9016 | 1 | 0.8815 | 1 | 0.6969 | 1 |
| SignSGD | 0.8663 | 32 | 0.5191 | 32 | 0.3955 | 32 |
| TernGrad | 0.8480 | 16 | 0.7418 | 16 | 0.6174 | 16 |
| QSGD (4 bits) | 0.8894 | 8 | 0.8545 | 8 | 0.6837 | 8 |
| QSGD (6 bits) | 0.9006 | 5.33 | 0.8803 | 5.33 | 0.6969 | 5.33 |
| Adaptive | 0.8991 | 6.53 | 0.8787 | 5.52 | 0.6943 | 5.93 |
| AdaQS | 0.9001 | 6.53 | 0.8809 | 5.35 | 0.6960 | 5.11 |
| DQSGD (Ours) | 0.8997 | **6.81** | 0.8793 | **7.11** | 0.6959 | **6.73** |

**Fixed Bits vs. Adaptive Bits**. Figure 1 shows the comparison results of fixed bit algorithm QSGD and our proposed DQSGD on CIFAR-10. Figure 1 (a) and Figure 1 (b) show the testing accuracy curves and the training loss curves, respectively. Figure 1 (c) shows the bits allocation of each iteration of DQSGD, and Figure 1 (d) represents the communication overhead used in the training process of different quantization schemes. From these results, we can see that although QSGD (2 bits) and QSGD (4 bits) have less communication cost, they suffer up to about $14\%$ and $2.7\%$ accuracy degradation compared with Vanilla SGD. The accuracy of QSGD (6 bits) and DQSGD is similar to that of Vanilla SGD, but the communication overhead of DQSGD is reduced up to $25\%$ compared with that of QSGD (6 bits). This shows that our dynamic quantization strategy can effectively reduce the communication cost compared with the fixed quantization scheme. Figure 2 shows the accuracy of QSGD and DQSGD under different compression ratios. It can be seen that DQSGD can achieve higher accuracy than QSGD under the same communication cost.

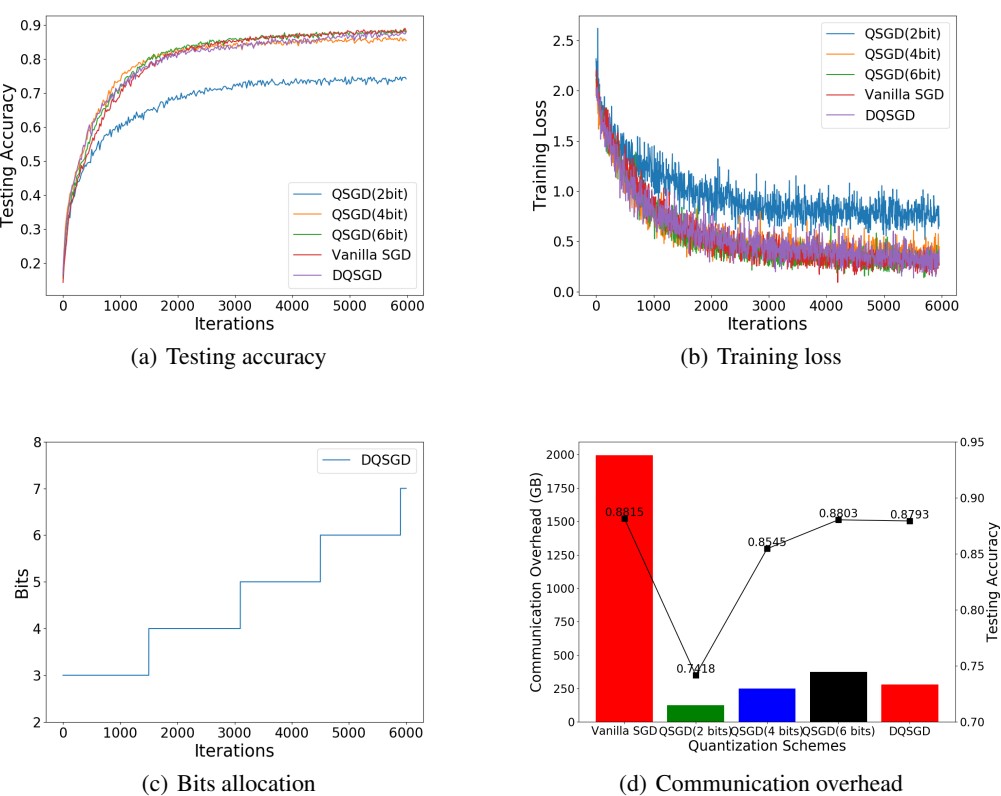

(a) Testing accuracy

(b) Training loss

(c) Bits allocation

(d) Communication overhead

Figure 1: The comparison results of QSGD and DQSGD on CIFAR-10.

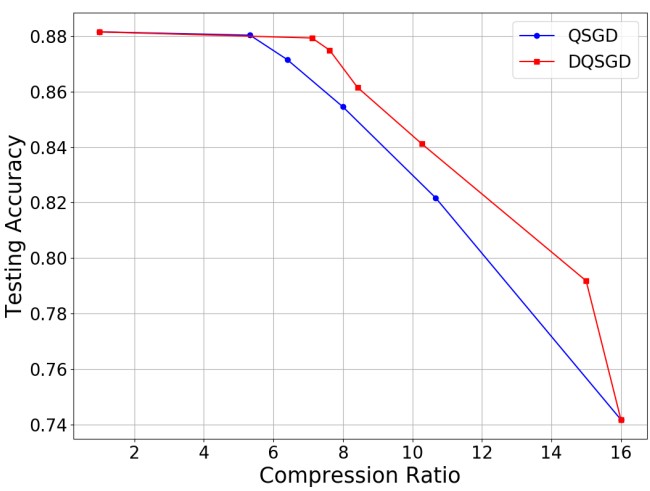

Figure 2: Testing accuracy of QSGD and DQSGD under different compression ratios on CIFAR-10.

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

## A   PROOF OF PROPOSITION 1

Suppose $\frac{g_i}{\|g\|_p} \sim U(\frac{l}{s}, \frac{l+1}{s})$ and let $\epsilon_i^{'} = \frac{g_i}{\|g\|_p} - \zeta(g_i, s)$, for $0 < \epsilon_0 < \frac{1}{s}$, we have:

$$p\{\epsilon_i^{'} = \epsilon_0\} = p\{\frac{g_i}{\|g\|_p} = \frac{l}{s} + \epsilon_0\} \cdot p\{\frac{l}{s} | \frac{g_i}{\|g\|_p} = \frac{l}{s} + \epsilon_0\}$$

$$= s \cdot \frac{\frac{1}{s} - \epsilon_0}{\frac{1}{s}}$$

$$= s - s^2\epsilon_0$$

Similarly, for $-\frac{1}{s} < \epsilon_0 < 0$, we have:

$$p\{\epsilon_i^{'} = \epsilon_0\} = p\{\frac{g_i}{\|g\|_p} = \frac{l+1}{s} + \epsilon_0\} \cdot p\{\frac{l+1}{s} | \frac{g_i}{\|g\|_p} = \frac{l+1}{s} + \epsilon_0\}$$

$$= s \cdot \frac{\frac{1}{s} + \epsilon_0}{\frac{1}{s}}$$

$$= s + s^2\epsilon_0$$

Considering that $Q_s(g_i) = \|g\|_p \cdot \text{sgn}(g_i) \cdot \zeta(g_i, s)$ and let $\hat{\epsilon}_i = g_i - Q_s(g_i)$, so we have:

$$p(\hat{\epsilon}_i) = \begin{cases} \frac{s}{\|g\|_p} - \frac{s^2}{\|g\|_p^2}\hat{\epsilon}_i & 0 < \hat{\epsilon}_i \le \frac{\|g\|_p}{s} \\ \frac{s}{\|g\|_p} + \frac{s^2}{\|g\|_p^2}\hat{\epsilon}_i & -\frac{\|g\|_p}{s} \le \hat{\epsilon}_i \le 0 \end{cases}$$

## B   PROOF OF THEOREM 1

Considering function $F$ is $L -$ smooth, and using Assumption 1, we have:

$$F(\mathbf{x}^{(n+1)}) \le F(\mathbf{x}^{(n)}) + \nabla F(\mathbf{x}^{(n)})^{\text{T}}(\mathbf{x}^{(n+1)} - \mathbf{x}^{(n)}) + \frac{L}{2}\|\mathbf{x}^{(n+1)} - \mathbf{x}^{(n)}\|_2^2$$

For QSGD, $\mathbf{x}^{(n+1)} = \mathbf{x}^{(n)} - \eta Q_{s_n}[g(\mathbf{x}^{(n)})]$, so:

$$F(\mathbf{x}^{(n+1)}) \le F(\mathbf{x}^{(n)}) + \nabla F(\mathbf{x}^{(n)})^{\text{T}}(-\eta Q_{s_n}[g(\mathbf{x}^{(n)})]) + \frac{L}{2}\|-\eta Q_{s_n}[g(\mathbf{x}^{(n)})]\|_2^2$$

$$= F(\mathbf{x}^{(n)}) - \eta \nabla F(\mathbf{x}^{(n)})^{\text{T}} Q_{s_n}[g(\mathbf{x}^{(n)})] + \frac{L\eta^2}{2}\|Q_{s_n}[g(\mathbf{x}^{(n)})]\|_2^2$$

Taking total expectations, and using Lemma 1, this yields:

$$\mathbb{E}[F(\mathbf{x}^{(n+1)})] \le F(\mathbf{x}^{(n)}) + (-\eta + \frac{L\eta^2}{2})\|\nabla F(\mathbf{x}^{(n)})\|_2^2 + \frac{L\eta^2\sigma^2}{2K} + \frac{L\eta^2 d}{12s_n^2}\|g(\mathbf{x}^{(n)})\|_p^2$$

Considering that function $F$ is $\mu - $ strongly convex, and using Assumption 2, so:

$$\mathbb{E}[F(\mathbf{x}^{(n+1)})] \leq F(\mathbf{x}^{(n)}) - (2\mu\eta - L\mu\eta^2)[F(\mathbf{x}^{(n)}) - F(\mathbf{x}^*)] + \frac{L\eta^2\sigma^2}{2K}$$
$$+ \frac{L\eta^2 d}{12 s_n^2}\|g(\mathbf{x}^{(n)})\|_p^2$$

Subtracting $F(\mathbf{x}^*)$ from both sides, and let $\alpha = 1 - 2\mu\eta + L\mu\eta^2$, so:

$$\mathbb{E}[F(\mathbf{x}^{(n+1)}) - F(\mathbf{x}^*)] \leq \alpha[F(\mathbf{x}^{(n)}) - F(\mathbf{x}^*)] + \frac{L\eta^2\sigma^2}{2K} + \frac{L\eta^2 d}{12 s_n^2}\|g(\mathbf{x}^{(n)})\|_p^2$$

Applying this recursively:

$$\mathbb{E}[F(\mathbf{x}^{(N)}) - F(\mathbf{x}^*)] \leq \alpha^N[F(\mathbf{x}^{(0)}) - F(\mathbf{x}^*)] + \frac{L\eta^2\sigma^2(1 - \alpha^N)}{2K(1-\alpha)}$$
$$+ \frac{Ld\eta^2}{12}\sum_{n=0}^{N-1}\alpha^{N-1-n}\frac{1}{s_n^2}\|g(\mathbf{x}^{(n)})\|_p^2$$

Similarly, let $\beta = 1 - 2L\eta + L\mu\eta^2$

$$\mathbb{E}[F(\mathbf{x}^{(N)}) - F(\mathbf{x}^*)] \geq \beta^N[F(\mathbf{x}^{(0)}) - F(\mathbf{x}^*)] + \frac{\mu\eta^2\sigma^2(1 - \beta^N)}{2K(1-\beta)}$$
$$+ \frac{\mu d\eta^2}{12}\sum_{n=0}^{N-1}\beta^{N-1-n}\frac{1}{s_n^2}\|g(\mathbf{x}^{(n)})\|_p^2$$

## C  PROOF OF THEOREM 2

Both SGD and QSGD can be considered a general kind of optimization dynamics, namely, gradient descent with unbiased noise. Based on the central limit theorem, it is assumed that the noise caused by sampling and quantization obeys Gaussian distribution, that is, $Q_{s_n}[g(\mathbf{x}^{(n)})] = \nabla F(\mathbf{x}^{(n)}) + \boldsymbol{\epsilon}^{(n)}, \boldsymbol{\epsilon}^{(n)} \sim \mathcal{N}(\mathbf{0}, \boldsymbol{\Sigma}(\mathbf{x}^{(n)}))$. Therefore, we can consider Equation (2) as the discrimination of the Gaussian process:

$$\mathbf{x}^{(n+1)} = \mathbf{x}^{(n)} - \eta\nabla F(\mathbf{x}^{(n)}) - \eta\boldsymbol{\epsilon}^{(n)}, \boldsymbol{\epsilon}^{(n)} \sim \mathcal{N}(\mathbf{0}, \boldsymbol{\Sigma}(\mathbf{x}^{(n)})) \tag{21}$$

The error for general Gaussian processes is hard to analyze due to the intractableness of the integrals, so we only consider a quadratic problem:

$$\mathbf{x}^{(n+1)} = \mathbf{x}^{(n)} - \eta\nabla F(\mathbf{x}^{(n)}) - \eta\boldsymbol{\epsilon}^{(n)}$$
$$= \mathbf{x}^{(n)} - \eta[\mathbf{H}\mathbf{x}^{(n)} + \mathbf{A}] - \eta\boldsymbol{\epsilon}^{(n)}$$
$$= (\mathbf{I} - \eta\mathbf{H})\mathbf{x}^{(n)} - \eta\mathbf{A} - \eta\boldsymbol{\epsilon}^{(n)}$$

Considering $\nabla F(\mathbf{x}^*) = \eta\mathbf{A} + \eta\mathbf{H}\mathbf{x}^* = 0$, subtracting $\mathbf{x}^*$ from both sides, and rearranging, this yields:

$$\mathbf{x}^{(n+1)} - \mathbf{x}^* = (\mathbf{I} - \eta\mathbf{H})\mathbf{x}^{(n)} - \eta\mathbf{A} - \mathbf{x}^* - \eta\boldsymbol{\epsilon}^{(n)}$$
$$= (\mathbf{I} - \eta\mathbf{H})(\mathbf{x}^{(n)} - \mathbf{x}^*) - \eta\mathbf{A} - \eta\mathbf{H}\mathbf{x}^* - \eta\boldsymbol{\epsilon}^{(n)}$$
$$= (\mathbf{I} - \eta\mathbf{H})(\mathbf{x}^{(n)} - \mathbf{x}^*) - \eta\boldsymbol{\epsilon}^{(n)}$$

Applying this recursively, let $\boldsymbol{\rho} = \mathbf{I} - \eta\mathbf{H}$, we have:

$$\mathbf{x}^{(N)} - \mathbf{x}^* = \boldsymbol{\rho}^N(\mathbf{x}^{(0)} - \mathbf{x}^*) - \sum_{n=0}^{N-1}[\eta\boldsymbol{\rho}^{N-1-n}\boldsymbol{\epsilon}^{(n)}]$$

Considering that $\boldsymbol{\epsilon}^{(n)} \sim \mathcal{N}(\mathbf{0}, \boldsymbol{\Sigma}(\mathbf{x}^{(n)}))$, then:

$$\sum_{n=0}^{N-1}[\eta\boldsymbol{\rho}^{N-1-n}\boldsymbol{\epsilon}^{(n)}] = \sum_{n=0}^{N-1}[\eta\boldsymbol{\rho}^{N-1-n}\boldsymbol{\Sigma}(\mathbf{x}^{(n)})^{\frac{1}{2}}\mathcal{N}(\mathbf{0}, \mathbf{I})]$$

$$= \sum_{n=0}^{N-1}\{\sqrt{\eta}\boldsymbol{\rho}^{N-1-n}\boldsymbol{\Sigma}(\mathbf{x}^{(n)})^{\frac{1}{2}}[\mathbf{W}(n+1) - \mathbf{W}(n)]\}$$

$$\equiv I(N)$$

where, $\mathbf{W}$ is a standard $d$ − dimensional Wiener process, and $I(N)$ is an Ito integral. Hence $\mathbf{x}^{(N)} = \mathbf{x}^* + \boldsymbol{\rho}^N(\mathbf{x}^{(0)} - \mathbf{x}^*) - I(N)$, then:

$$F(\mathbf{x}^{(N)}) = \frac{1}{2}\mathbf{x}^{(N)^\mathrm{T}}\mathbf{H}\mathbf{x}^{(N)} + \mathbf{A}^\mathrm{T}\mathbf{x}^{(N)} + B$$

$$= \frac{1}{2}(\mathbf{x}^{(0)} - \mathbf{x}^*)^\mathrm{T}(\boldsymbol{\rho}^N)^T\mathbf{H}\boldsymbol{\rho}^N(\mathbf{x}^{(0)} - \mathbf{x}^*) + \frac{1}{2}I(N)^\mathrm{T}\mathbf{H}I(N)$$

$$- [\boldsymbol{\rho}^N(\mathbf{x}^{(0)} - \mathbf{x}^*) + \mathbf{x}^* + \mathbf{A}]^\mathrm{T}\mathbf{H}I(N) + F(\mathbf{x}^*)$$

Subtracting $F(\mathbf{x}^*)$ from both sides, taking total expectations, and rearranging, this yields:

$$\mathbb{E}[F(\mathbf{x}^{(N)}) - F(\mathbf{x}^*)] = \frac{1}{2}(\mathbf{x}^{(0)} - \mathbf{x}^*)^\mathrm{T}(\boldsymbol{\rho}^N)^\mathrm{T}\mathbf{H}\boldsymbol{\rho}^N(\mathbf{x}^{(0)} - \mathbf{x}^*) + \frac{1}{2}\mathbb{E}[I(N)^\mathrm{T}\mathbf{H}I(N)]$$

$$- [\boldsymbol{\rho}^N(\mathbf{x}^{(0)} - \mathbf{x}^*) + \mathbf{x}^* + \mathbf{A}]^\mathrm{T}\mathbf{H}\mathbb{E}[I(N)]$$

The property of Ito integral $I(N)$ is:

$$\mathbb{E}[I(N)] = 0$$

$$\mathbb{E}[I(N)^\mathrm{T}\mathbf{H}I(N)] = \sum_{n=0}^{N-1}\eta^2\mathrm{Tr}[\boldsymbol{\rho}^{N-1-n}\boldsymbol{\Sigma}(\mathbf{x}^{(n)})\mathbf{H}(\boldsymbol{\rho}^{N-1-n})^\mathrm{T}]$$

Using this property, we have:

$$\mathbb{E}[F(\mathbf{x}^{(N)}) - F(\mathbf{x}^*)] = \frac{1}{2}(\mathbf{x}^{(0)} - \mathbf{x}^*)^\mathrm{T}(\boldsymbol{\rho}^N)^\mathrm{T}\mathbf{H}\boldsymbol{\rho}^N(\mathbf{x}^{(0)} - \mathbf{x}^*)$$

$$+ \frac{\eta^2}{2}\sum_{n=0}^{N-1}\mathrm{Tr}[\boldsymbol{\rho}^{N-1-n}\boldsymbol{\Sigma}(\mathbf{x}^{(n)})\mathbf{H}(\boldsymbol{\rho}^{N-1-n})^\mathrm{T}]$$

## D    PROOF OF THEOREM 3

Considering function $F$ is $L$ − smooth, using the result of Appendix B, we have:

$$\mathbb{E}[F(\mathbf{x}^{(n+1)})] \le F(\mathbf{x}^{(n)}) + (-\eta + \frac{L\eta^2}{2})\|\nabla F(\mathbf{x}^{(n)})\|_2^2 + \frac{L\eta^2\sigma^2}{2K} + \frac{L\eta^2 d}{12s_n^2}\|g(\mathbf{x}^{(n)})\|_p^2$$

Subtracting $F(\mathbf{x}^{(n)})$ from both sides, then applying it recursively, this yields:

$$\mathbb{E}[F(\mathbf{x}^{(N)}) - F(\mathbf{x}^{(0)})] \leq -(\eta - \frac{L\eta^2}{2}) \sum_{n=0}^{N-1} \mathbb{E}[\|\nabla F(\mathbf{x}^{(n)})\|_2^2] + \frac{LN\eta^2\sigma^2}{2K}$$

$$+ \frac{Ld\eta^2}{12} \sum_{n=0}^{N-1} \frac{1}{s_n^2}\|g(\mathbf{x}^{(n)})\|_p^2$$

Considering that $F(\mathbf{x}^{(N)}) \geq F(\mathbf{x}^*)$, so:

$$\frac{1}{N} \sum_{n=0}^{N-1} \mathbb{E}[\|\nabla F(\mathbf{x}^{(n)})\|_2^2] \leq \frac{2}{2N\eta - LN\eta^2}[F(\mathbf{x}^{(0)}) - F(\mathbf{x}^*)] + \frac{L\eta\sigma^2}{(2 - L\eta)K}$$

$$+ \frac{Ld\eta}{6(2 - L\eta)N} \sum_{n=0}^{N-1} \frac{1}{s_n^2}\|g(\mathbf{x}^{(n)})\|_p^2$$

## E  ALGORITHM

---
**Algorithm 1** Dynamic quantized SGD
---
1: **Input:** Learning rate $\eta$, Initial point $\mathbf{x}^{(0)} \in \mathbb{R}^d$, Hyperparametric $k$, $\alpha$

2: **for** each iteration $n = 0, 1, ..., N - 1$: **do**

3:     $g(\mathbf{x}^{(n)}) \leftarrow$ compute gradient of a batch of data

4:     $\|g(\mathbf{x}^{(n)})\| \leftarrow$ calculate the norm of $g(\mathbf{x}^{(n)})$

5:     $B_n \leftarrow$ determine the quantization bits

6:     $\tilde{g}(\mathbf{x}^{(n)}) \leftarrow$ quantize $(g(\mathbf{x}^{(n)}), B_n)$

7:     Update the parameter: $\mathbf{x}^{(n+1)} = \mathbf{x}^{(n)} - \eta\tilde{g}(\mathbf{x}^{(n)})$

8: **end for**

---

We make an assumptions as follows.

**Assumption 4.** *(Second moment bound). If $F(\mathbf{x})$ is $L - smooth$, so the $l_p$ norm of the minibach stochastic gradient $g(\mathbf{x})$ satisfied:*

$$\|g(\mathbf{x})\|_p^2 \leq 2L[F(\mathbf{x}) - F(\mathbf{x}^*)]^\gamma \tag{22}$$

It is noted that Assumption 4 is a generalization of Equation (7).

Based on this quantization scheme 18 and Assumption 4, we can get the quantization error:

$$\delta'_{\text{DQSGD}} \leq \frac{L^2\eta^2 d\alpha^{N-1}[F(\mathbf{x}^{(0)}) - F(\mathbf{x}^*)]^\gamma}{6 \times 4^{(C-32N-dN)/dN}} N\alpha^{(\gamma-1)(N-1)/2} \tag{23}$$

Accordingly, if we fix the bits, the quantization error is:

$$\delta'_{\text{Fixed}} \leq \frac{L^2\eta^2 d\alpha^{N-1}[F(\mathbf{x}^{(0)}) - F(\mathbf{x}^*)]^\gamma}{6 \times 4^{(C-32N-dN)/dN}} \sum_{n=0}^{N-1} \alpha^{(\gamma-1)n} \tag{24}$$

Comparing (23) and (24), we can see that our scheme reduces the error bound about:

$$\delta'_{\text{Fixed}} - \delta'_{\text{DQSGD}} \approx \frac{L^2\eta^2 d\alpha^{N-1}[F(\mathbf{x}^{(0)}) - F(\mathbf{x}^*)]^\gamma}{6 \times 4^{(C-32N-dN)/dN}}\lambda_1 \tag{25}$$

---

**Algorithm 2** Dynamic QSGD in Distributed Learning

---

1: **Input:** Learning rate $\eta$, Initial point $\mathbf{x}^{(0)} \in \mathbb{R}^d$, Hyperparametric $k$, $\alpha$

2: **for** each iteration $n = 0, 1, ..., N - 1$: **do**

3:     **On each worker** $l = 1, ..., W$**:**

4:     $g^l(\mathbf{x}^{(n)}) \leftarrow$ compute gradient w.r.t. a batch of data

5:     $\tilde{g}^l(\mathbf{x}^{(n)}) \leftarrow$ quantize $(g^l(\mathbf{x}^{(n)}), B_n)$

6:     send $\tilde{g}^l(\mathbf{x}^{(n)})$ to server

7:     receive $g(\mathbf{x}^{(n)})$ and $B_{n+1}$ from server

8:     **On server:**

9:     collect all $W$ gradients $\tilde{g}^l(\mathbf{x}^{(n)})$ from workers

10:     average: $\tilde{G}(\mathbf{x}^{(n)}) = \frac{1}{W} \sum_{l=1}^{W} \tilde{g}^l(\mathbf{x}^{(n)})$

11:     $\|\tilde{G}(\mathbf{x}^{(n)})\| \leftarrow$ calculate the norm of $g(\mathbf{x}^{(n)})$

12:     $B_{n+1} \leftarrow$ Determine the quantization bits for the next iteration $(\|\tilde{G}(\mathbf{x}^{(n)})\|)$

13:     send $\tilde{G}(\mathbf{x}^{(n)})$ and $B_{n+1}$ to all workers

14: **end for**

---

where $\lambda_1 = \sum_{n=0}^{N-1} \alpha^{(\gamma-1)n} - N\alpha^{(\gamma-1)(N-1)/2}$ is the difference between arithmetic mean and geometric mean. When $\gamma = 1$, $\lambda_1 = 0$. If $\gamma \neq 1$, $\lambda_1 > 0$.

Based on quantization scheme (20) and Assumption 4, the quantization error is:

$$\delta_{\text{DQSGD}}''' \leq \frac{L^2 \eta d [F(\mathbf{x}^{(0)}) - F(\mathbf{x}^*)]^\gamma}{3N(2 - L\eta) \times 4^{(C-32N-dN)/dN}} N\alpha^{\gamma(N-1)/2} \tag{26}$$

Accordingly, if we fix the bits, the quantization error is:

$$\delta_{\text{Fixed}}''' \leq \frac{L^2 \eta d [F(\mathbf{x}^{(0)}) - F(\mathbf{x}^*)]^\gamma}{3N(2 - L\eta) \times 4^{(C-32N-dN)/dN}} \sum_{n=0}^{N-1} \alpha^{\gamma n} \tag{27}$$

Comparing (26) and (27), we can see that our scheme reduces the error bound about:

$$\delta_{\text{Fixed}}''' - \delta_{\text{DQSGD}}''' \approx \frac{L^2 \eta d [F(\mathbf{x}^{(0)}) - F(\mathbf{x}^*)]^\gamma}{3N(2 - L\eta) \times 4^{(C-32N-dN)/dN}} \lambda_2 \tag{28}$$

where $\lambda_2 = \sum_{n=0}^{N-1} \alpha^{\gamma n} - N\alpha^{\gamma(N-1)/2}$ is the difference between arithmetic mean and geometric mean. Consider that $\gamma \neq 0$, so $\lambda_2 > 0$.

## F    EXPERIMENTS

### F.1    DATASETS AND BASELINE

We evaluate our method DQSGD on three datasets: AG-News, CIFAR-10, and CIFAR-100. AG-News dataset (Zhang et al., 2015) contains four categorized news articles, and the number of training samples for each class is 30000 and testing 1900. CIFAR-10/100 (Krizhevsky et al., 2009) dataset are all contain 60,000 $32 \times 32$ RGB images, which are divided into 10 and 100 classes, respectively. We compare DQSGD with the following gradients quantization methods:

- SignSGD (Seide et al., 2014): To take the sign of each coordinate of the stochastic gradient vector.

- TernGrad (Wen et al., 2017): Quantizes gradients to ternary levels $\{-1; 0; 1\}$.

• QSGD (Alistarh et al., 2017): It is a family of compression schemes. The specific quantization operation is shown in equation 3. In our experiments, we replace the $\|g\|_2$ in the original text with $\|g\|_\infty$.

• Adaptive (Oland & Raj, 2015): This dynamic scheme considers that for the gradient with larger root-mean-squared (RMS) value, more quantization bits are used.

• AdaQS (Guo et al., 2020): It is an adaptive quantization scheme that using few quantization bits in the early epochs and gradually increase bits in the later epochs.

Table 2: Baselines

|  | Unbiased | Basis for determining bits |
|---|---|---|
| SignSGD (Seide et al., 2014) | No | Fixed bits |
| TernGrad (Wen et al., 2017) | Yes | Fixed bits |
| QSGD (Alistarh et al., 2017) | Yes | Fixed bits |
| Adaptive (Oland & Raj, 2015) | Yes | Gradient's root-mean-squared value |
| AdaQS (Guo et al., 2020) | Yes | Gradient's mean to standard deviation ratio, Iteration number |

## F.2 EXPERIMENTAL SETUP

We conduct simulations for $W = 8$ workers. For AG-News, we use 300-dim embeddings pre-trained on Glove; then, each word can be further encoded sequentially using two layers bidirectional LSTM (BiLSTM). Furthermore, we use the self-attention mechanism to obtain the sentence embedding. The classifier is two fully connected layers of size 128 and 4 neurons, respectively. We training CIFAR-10 on Resnet18 (He et al., 2016) and CIFAR-100 on Resnet34 (He et al., 2016), respectively. Other parameters information is shown in Table 3. All results were the average of four random runs.

Table 3: Parameters

| Dataset | Net | Learning rate | Batchsize | Interations | Hyperparameters in DQSGD |
|---|---|---|---|---|---|
| AG-News | BiLSTM | 0.005 | 32 | 1000 | $k = 5, \alpha = 0.994$ |
| CIFAR-10 | Resnet18 | 0.1 | 32 | 6000 | $k = 20, \alpha = 0.999$ |
| CIFAR-100 | Resnet34 | 0.01 | 64 | 6000 | $k = 10, \alpha = 0.999$ |

