# OpenReview forum: "DQSGD: DYNAMIC QUANTIZED STOCHASTIC GRADIENT DESCENT FOR COMMUNICATION-EFFICIENT DISTRIBUTED LEARNING"
_ICLR.cc/2021/Conference — Reject_

### Official Review · AnonReviewer4 · 2020-10-21
**The paper proposes Dynamic Quantized SGD (DQSGD) algorithm for distributed learning. It shows some convergence properties of this algorithm and provides some experiments to assess its efficiency.**

**Rating:** 2
**Confidence:** 4

**Review:**

-It is known that Assumption 3 equation (10) (bounded variance) with Assumption 2 (strong convexity) leads to a contradiction. Thus having these two assumptions together is strong.
There are some recent works that overcome Assumption 3 by a different assumption known as the expected smoothness like in the following work: Gower,  R.  M.,  Richtarik,  P.,  and  Bach,  F.    Stochastic quasi-gradient methods: Variance reduction via Jacobian sketching.arxiv:1805.02632, 2018.
The authors may revise their strongly convex results part using this kind of assumption.

-In proposition 1 equation (13), it is not clear what the authors mean by the probability of a random variable. I checked the proof but I did not understand it either.

-Some reported results are already known in the literature and the paper gives the impression that these results are new, especially that the authors give the proofs.
Examples of such results: the unbiasedness of the quantization, its bounded variance, and the first part of theorem 1...

-Page 5, concerning the quadratic example: this is a trivial case and the only case where one can hope the lower bound to match the upper bound. In fact, alpha = beta iff L=mu and from Assumptions 1 & 2 we get that F is quadratic with mu=L which implies H = mu I.

- Equation (20): for me this one of the main results of the paper. But I did not see its proof anywhere?

---

> ### Author Response · Authors · 2020-11-24
> **Reply to AnonReviewer4**
>
> A1: In ["SGD and Hogwild! Convergence Without the Bounded Gradients Assumption" by Nguyen et al.], the author proves that the strong convexity assumption is contradictory to the assumption of uniform boundedness of stochastic gradient. To our knowledge, there is no work to prove that (bounded variance) and (strong convexity) are contradictory. "Variance bound" only gives the boundary of the stochastic gradient variance, and the stochastic gradient itself can be infinite. And 'uniform boundedness' gives the upper bound of the second moments of the stochastic gradient, so the stochastic gradient is also bounded.
>
> A2: Proposition 1 gives the probability distribution of the quantization noise. Then we can get the variance of quantized stochastic gradient.
>
> A3: In proposition 1, we obtain a tighter variance bound of a quantized stochastic gradient than that in the QSGD paper. For theorem 1, we hope to obtain the dynamic quantization scheme by minimizing the upper bound of the convergence error of QSGD, so we give the proof process. The form obtained is different from that in the QSGD paper.
>
> A4: We get the upper and lower bound of the general strongly convex problem and prove that the upper and lower bound is tight in special cases.

---

### Official Review · AnonReviewer1 · 2020-10-28
**Moderately novel, but inconsistent assumptions and inaccurate claims**

**Rating:** 4
**Confidence:** 5

**Review:**

The paper considers distributed learning using SGD and aims at improving the convergence rate of the quantized SGD. To this purpose, it theoretically characterizes the trade-off between communication cost and the model accuracy in training with quantized stochastic gradients.
Based on the theoretical analysis, the authors proposed a dynamic quantized SGD framework to optimize the quantization strategy to be analyzing the trade-off between communication and model error.
In addition to the theoretical analysis, the performance of the proposed communication scheme is evaluated extensively in computer vision tasks on CIFAR10/100, and NLP on AG-News dataset, and is shown to outperform the other considered quantization techniques.


Strengths:
Although the idea of changing the number of quantization bit during training has been proposed and studied before, but this paper looks at it from a new angle and by analyzing the convergence rate under different assumptions for the loss function, it computes the required number of bits to minimize the upper bound of the convergence error.

Weaknesses and questions:
1. In the proof of proposition 1, it is explicitly assumed that the SGs are uniformly distributed in each quantization bin. Although this might be true for sufficiently small quantization bins (large bits), for low bits (which is the interested region in distributed training) this assumption is not valid. This should be mentioned in the body of the claims (although this assumption is not true in general). To be more accurate, the distribution of error for example for $e>0$ can be written as $P(e)=(1-se) \sum_l P(g_i/\|g\|_p = e+l/s)$, ...
2. Theorem 2 assumes Gaussian noise, while the quantization noise is non-Gaussian, and prop. 1 assumes it is uniform. It seems that this assumption is not in line with the previous assumptions and practical cases.
3. The experiments section is not satisfactory. The QSGD and TernGrad methods are relatively old and better quantization methods based on dithering, transforming, ... has already been proposed. Compared to Adaptive and AdaQS, both these methods achieve higher accuracy with a lower compression ratio. On the other hand, QSGD(4 bits) with a higher compression ratio achieves a comparable accuracy. Moreover, the convergence rate plots (Fig. 1) shows that all methods converge almost at the same rate, despite the theoretical analysis and the motivation of the dynamic bt allocation
For a fair comparison, the compression ratio of all methods should be fixed to the same value and the convergence rate as in Fig. 1(a),(b) (in addition to the final accuracy) should be evaluated. The limited experiments and not appropriate comparison setups are major shortcomings of the paper.

Minor:
Theorem 1 uses assumption 3, not stated in the body.

---

> ### Author Response · Authors · 2020-11-24
> **Reply to AnonReviewer1**
>
> A1: If we remove this assumption, we can get the quantized stochastic gradient variance's upper bound. Based on this upper bound, we can still get the same quantization scheme.
>
> A2: In the proof of Theorem 1, we find that the convergence error due to quantization is only related to the variance of quantization noise and has nothing to do with the specific distribution of noise. Therefore, we can reasonably approximate the quantization noise to Gaussian noise with the same variance.
>
> A3:In Table 1, we compare the testing accuracy and communication compression ratio of different algorithms under different tasks. The baselines included the fixed bit algorithms (SignSGD, TernGrad, QSGD) and dynamic bits algorithms (Adaptive, AdaQS). In Fig.1 and Fig.2, we give the training process of our algorithm and the tradeoff between model accuracy and communication overhead.

---

### Official Review · AnonReviewer2 · 2020-10-28
**Official Blind Review #2**

**Rating:** 4
**Confidence:** 4

**Review:**

This paper proposed an adaptive quantized method which is derived by minimizing a constrained quantization error bound. The theoretical analysis suggests adjusting the quantization level according to the gradient norm, convergence rate of the model, and the current iteration number. Theoretical results show that the dynamic bits leads to better error bound than the fixed bits. The result is intuitive. Overall, the paper is clearly written. But the improvement is not significant enough to warrant a publication at ICLR.

1. In the proof of Proposition 1, it is assumed that $\frac{g_i }{ \|g\|_p} \in U(\frac{l}{s}, \frac{l+1}{s})$. Is this empirically supported? And, the final result looks wrong to me. For example, $p(\frac{g_i}{\|g\|_p} = \frac{l}{s} + \epsilon_0)$ should be $1/s$ and $s - s^2\epsilon_0$ can be much larger than 1.

2. B_n in (18) depends on \alpha and k. How can we estimate them in practice? On the other hand, (18) shows that more bits may need to be used in the later stage of training (due to exponent $N - n$), while (20) suggests a smaller number of bits in later iterations (as gradient norm typically has a trend of decreasing during training). They are contradictory to each other.

3. In the experiment, (18) or (20) is used?

4. As the proposed method uses dynamic number of bits, how do we compute its compression ratio? Is it averaged compression ratio across iterations.

5. From Table 1, we can see that though the proposed method has slightly higher compression ratio than QSGD, its accuracy is worse. Since the compression overhead is nontrivial in practice and there is no report on training time, it is not clear if the proposed method is faster in terms of CPU wall-clock time.

6. The baseline looks weak. There are a number of papers showing that error feedback can fix the poor performance of signSGD (Karimireddy 2019; Tang 2019; Zheng 2019). And, is momentum used in the experiment?

Karimireddy, Sai Praneeth, et al. "Error feedback fixes signsgd and other gradient compression schemes." ICML 2019.

Tang, Hanlin, et al. "Doublesqueeze: Parallel stochastic gradient descent with double-pass error-compensated compression." ICML 2019.

Zheng, S., Huang, Z., Kwok, J. "Communication-efficient distributed blockwise momentum SGD with error-feedback." NeurIPS 2019.

---

> ### Author Response · Authors · 2020-11-24
> **Reply to Official Blind Review #2**
>
> A1: Quantization level is roughly exponential to the number of quantized bits, so for small quantization bins, this assumption is reasonable. Furthermore, Proposition 1 is also empirically supported. $p({g_i}/{\|g\|_p} = l/s+\epsilon_0)$ should be $s$. For uniform distribution, the probability density is the reciprocal of the interval length. $s-s^2 \epsilon_0$ is the probability density and can be greater than 1.
>
> A2: k depends on the total communication budget C, and $\alpha=\alpha(L, \mu, \eta)$ is related to the convergence rate of the model. The larger the total communication budget C is, the greater k is; the faster the model's convergence rate is, the smaller $\alpha$ is. In our algorithm, k and $\alpha$ are hyper-parameter. Under the strongly convex assumption, we add a weight variable $\alpha^{N-n}$ to the gradient norm. It can be seen that with the training going on, this weight parameter will become larger and larger, which will reduce the quantization variance. So this is also a kind of variance reduction techniques. The quantization bits is mainly determined by $\alpha^{N-n}$ and the gradient norm.
>
> A3: We use Eq.(18) in the experiment.
>
> A4: Yes. The compression ratio is expressed as the division of communication overhead of Vanilla SGD by communication overhead
> of other quantization algorithms.
>
> A5: Fig.2 shows the accuracy of QSGD and DQSGD under different compression ratios. When the accuracy of QSGD and DQSGD is similar to that of Vanilla SGD, the communication overhead of DQSGD is reduced up to $25\%$ compared with that of QSGD (Fig. 1). It can be seen that DQSGD can achieve higher accuracy than QSGD under the same communication cost.
>
> A6: The baselines included the fixed bit algorithms (SignSGD, TernGrad, QSGD) and dynamic bits algorithms (Adaptive, AdaQS). This work's main contribution is to propose a novel framework to characterize the trade-off between communication cost and modeling error by dynamically quantizing gradients in distributed learning. So we need to keep the baseline under the same conditions. In future work, error compensation can be considered based on this work.

---

### Official Review · AnonReviewer3 · 2020-10-28
**This paper proposes a dynamic quantized SGD framework that finds the trade-off between communicated bits in quantized gradients and DNN model accuracy and dynamically determine the quantization level.**

**Rating:** 2
**Confidence:** 5

**Review:**

1.	The authors considered uniform upper bound of the stochastic gradients g_i. The authors may argue that "The classical theoretical analysis of SGD assumes that the stochastic gradients are uniformly bounded". But one can even strongly argue that this bound is actually $\infty$. Moreover, an even stronger argument can be made that the above assumption is in contrast with strong convexity. Please see ["SGD and Hogwild! Convergence Without the Bounded Gradients Assumption" by Nguyen et al.] as one of the instances. Please understand there are relaxed assumptions such as Strong growth condition on a stochastic gradient as in Assumption 4 of [1].
2.	There is work [2] that proposes we propose a flexible framework which adapts the compression level to the true gradient at each iteration, maximizing the improvement in the objective function that is achieved per communicated bit. How is work different from theirs? I suggest the authors mention this work and make connections with their present results.
3.	Please check Theorem 3.4. and 3.5 in QSGD paper. There is a count of bits communicating in each round for QSGD which depends on s, the quantization level. Moreover, you also mentioned that “quantization level is roughly exponential to the number of quantized bits”. In light of your formulation how the results in QSGD paper are relevant? Additionally, in Section 3, equations (3) and (4) are not your contributions. Therefore, please adequately cite their sources.
4.	What is the idea behind Proposition 1?
5.	What is the point of Theorem 2? On the other hand, where are the derivations/proofs of Equations (18) and (20)? Those are the main results of this paper if I am not wrong.
6.	In terms of experimental results, I had a hard time interpreting Table 1 and associated text in the paragraph “Test Accuracy vs Compression Ratio”. Especially, I do not know how do I understand the last sentence in the abovementioned paragraph. To do proper experiments by using compression techniques, the authors can check a very elaborative work and codebase by [Hang Xu et. al, Compressed Communication for Distributed Deep Learning: Survey and Quantitative Evaluation]. In that case, I would encourage the authors to plot relative data-volume vs. test accuracy similar to Figures 6 and 7 therein. In the present papers, the experiments and their presentations are substandard.

Additional comments:
1.	“Recently, with the booming”---Please refrain from using these types of words. You are preparing a scientific document, not a sci-fi novel.
2.	I have some reservations on this statement: “Existing algorithms often quantize parameters into a fixed number of bits, which is shown to be inefficient in balancing the communication-convergence trade-off (Seide et al., 2014; Alistarh et al., 2017; Bernstein et al., 2018). “ In my opinion, QSGD is robust and stable, For 1-bit SGD (both Seide and Bernstein), the proposed solution is to use error-feedback.
3.	“show good performance in some certain tasks”---bad sentence.



[1] Dutta et al. AAAI 2020, On the Discrepancy between the Theoretical Analysis and Practical Implementations of Compressed Communication for Distributed Deep Learning.
[2] Khirirat et al., 2020, A flexible framework for communication-efficient machine learning: from HPC to IoT.

---

> ### Author Response · Authors · 2020-11-24
> **Reply to AnonReviewer3**
>
> A1: We use the "Variance bound" assumption, which is different from the assumption of the stochastic gradient's uniform boundedness. "Variance bound" only gives the boundary of the variance of the stochastic gradient, and the stochastic gradient itself can be infinite. And ‘uniform boundedness’ is the upper bound of the second moments of the stochastic gradient, so the stochastic gradient is also bounded.
>
> A2:  The work [2] models how the communication cost depends on the compression level for IoT.
> Our dynamic quantization algorithm can adjust the number of quantization bits adaptively by taking into account the norm of gradients, the communication budget, and the remaining number of iterations.
>
> A3: The Theorem 3.4. and 3.5 in QSGD paper fixed the number of quantization bits in the training process. In our work, we allow the quantization bits of different iterations to be different.
>
> A4:  Proposition 1 gives the probability distribution of the quantization noise. Then, we can obtain the variance bound of quantized stochastic gradient. In Theorem 1, we can see that the convergence error is directly related to the quantized stochastic gradient variance.
>
> A5: In Theorem 2, we give exact convergence errors for quadratic problems. Since the optimization problem in this paper is convex and can be solved directly by the Lagrange multiplier method, we omitted the proof in the paper.
>
> A6: In Table 1, we compare the testing accuracy and communication compression ratio of different algorithms under different tasks. The baselines included the fixed bit algorithms (SignSGD, TernGrad, QSGD) and dynamic bits algorithms (Adaptive, AdaQS). In Fig. 1 and Fig.2, we give the training process of our algorithm and the tradeoff between model accuracy and communication overhead.

---

### Decision · Program_Chairs · 2021-01-07
**Final Decision**

**Decision:**

Reject

**Comment:**

The reviewers have a strong consensus towards rejection here, and I agree with this consensus , although I think some of the reviewers' concerns are misplaced. For example, the paper does not appear to use a magnitude upper bound that would be vacuous together with a strong convexity assumption (although variance bounds + strong convexity do cover only a small fraction of strongly convex learning tasks, these assumptions aren't vacuous).  Some feedback I have that perhaps was not covered by the reviewers:

Pros:

 - Studying the setting where the number of bits varies dynamically is very interesting (although, as Reviewer 3 points out, not entirely novel). There is significant possibility for improvement from this method, and your theory seems to back this up.

Cons:

 - The experimental setup is weak, and is measuring the wrong thing. When we run SGD to train a model, what we really care about is when the training finishes: the total wall clock time to train on some system. For compression methods with fixed compression rates, it's fine to use the number of bits transmitted as a proxy, because (when the number of bits transmitted is uniform over time) this will be monotonic in the wall-clock time. However, when the bits transmitted per iteration can change over time, this can have a difficult-to-predict effect on the wall-clock time, because of the potential for overlap between communication and computation (where below a certain number of bits sent, the system is not communication-bound). Wall-clock time experiments comparing against other more modern compression methods would significantly improve this paper.